

PeerJ Hubs
Published on behalf of

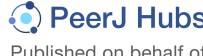


# A decade of study on the condition of western Cuban coral reefs, with low human impact

Hansel Caballero-Aragón[1], Susana Perera-Valderrama[1], Dorka Cobián-Rojas[2], Zaimiuri Hernández Gonzalez[3], Juliett González Méndez[4] and Elena De la Guardia[3]

[1] National Commission for the Knowledge and Use of Biodiversity, México City, México
[2] Guanahacabibes National Park, Sandino, Pinar del Río, Cuba
[3] Cayos de San Felipe National Park, La Coloma, Pinar del Río, Cuba
[4] National Center of Protected Areas, La Habana, Cuba

Corresponding author
Susana Perera-Valderrama,
sperera@conabio.gob.mx

## ABSTRACT

**Background.** The long-time study of coral reefs with low human impacts can provide information on the effects of regional pressures like climate change, and is an opportunity to document how these pressures are reflected in coral communities. An example of minimal local anthropogenic impacts are the Guanahacabibes coral reefs, located in the westernmost region of Cuba. The objectives of this study were: to evaluate the temporal variability of six benthic biological indicators of coral reefs, and to explore the possible relationship between predictive abiotic variables and biological response variables.

**Methods.** Four coral reef sites were sampled between 2008 and 2017, to analyze biological indicators (living coral cover, fleshy algae index, coral species richness, coral species abundance, coral trait groups species abundance, Functional Reef Index). Seven abiotic variables (wave exposure, sea surface temperature, degree heating week, chlorophyll-a concentration, particulate organic carbon, photosynthetically available radiation, and the diffuse attenuation coefficient) were compiled between 2007 and 2016, from remote sensing datasets, to analyze their relationship with the biological indicators. Permanova statistical analysis was used to evaluate trends in biological variables between sites and years, and Routine Analysis Based on Linear Distances (DISTLM) was used to explore some dependencies between biotic and abiotic variables.

**Results.** We found significant variability in the temporal analysis, with a decrease in living coral cover, a decline in the predominance of the branching and massive framework reef-building species, a decline in *Orbicella* species abundance, and an increase in the fleshy algae index. Some abiotic variables (average of degree heating weeks, standard deviation of the diffuse attenuation coefficient, average of the sea surface temperature, among others) significantly explained the variability of biological indicators; however, determination coefficients were low.

**Conclusions.** Certain decrease in the functionality of the coral reef was appreciated, taking into account the predominance of secondary and nom-massive framework reef-building species in the last years. A weak association between abiotic and biological variables was found in the temporal analysis. The current scenario of the condition of the coral reefs seems to be regulated by the global effects of climate change, weakly associated effects, and in longer terms.

## INTRODUCTION

Diverse suites of threats are modifying the structure and function of marine ecosystems, and very few ecosystems currently resemble their 'natural' state (*Rogers et al., 2015*; *Cramer et al., 2021*). Anthropogenic stressors threaten more than the 60% of the world's coral reefs (*Burke et al., 2011*), including the dumping of large sediment loads into the sea, organic and inorganic pollution (*Gove et al., 2015*), physical damage, and overfishing (*Edwards et al., 2014*). Many of these impacts are associated with increases in human settlements, coastal development, tourism and agriculture (*Williams et al., 2019*; *Cramer et al., 2020*).

The coral reefs condition in the Caribbean has declined because of the acute and direct effects of human activities, including eutrophication and dredging. These disturbances have produced a high decrease in coral cover, accompanied by changes in the relative abundance of coral species, reduced species diversity, and a shift to slower-growing coral species which is directly related to the tolerance of individual species to sediment stress (*Larsen & Webb, 2009*; *Ramos-Scharrón, Torres-Pulliza & Hernández-Delgado, 2015*; *Takesue et al., 2021*). The negative effects on corals are exacerbated by stresses from global changes, such as bleaching in response to rising sea temperatures (*Hughes et al., 2018a*; *Hughes et al., 2018b*), reduced calcification in organisms due to ocean acidification (*Albright et al., 2016*; *Courtney et al., 2020*), and more frequent storm damage (*Manfrino et al., 2013*).

The proximity of many reefs to human populations makes it difficult to separate the effects of local anthropogenic pressures from those of regional or global pressures (*Alcolado et al., 2010*). There are studies on reef extensions far from human population centers in the Caribbean (*e.g.*, *Sanvicente-Añorve et al., 2014*; *Aguilar-Perera et al., 2018*; *Ibarra-García et al., 2022*), but as a consequence of their remoteness, they are less represented in the literature than reefs close to human settlements. The study of remote reefs can provide useful information on the effects of regional and global pressures and represents a unique opportunity to document whether the degradation observed in reefs close to coastal population centers is reflected in more remote areas (*Alcolado et al., 2010*).

An example is the Guanahacabibes Peninsula, located in the westernmost region of Cuba. The area is included inside a National Park with restricted access and low anthropogenic exploitation (*Espinosa et al., 2012*). This area has been studied since the end of the 20th century (*e.g.*, *Alcolado et al., 2003*; *Guardia, Valdivia & González-Díaz, 2004*; *Caballero-Aragón et al., 2007*; *Perera-Valderrama et al., 2013*); however, a continuous study evaluating possible changes in the structure and condition of its benthic community has not yet been conducted.

Monitoring a single location over time can provide an early warning system of coral reefs stress, helping to diagnose possible causes of their degradation or recovery and to determine the best management and protection measures (*Mumby et al., 2014*). In this context, the results of a decade of study at four coral reef sites in Guanahacabibes are presented, with the objectives of: (i) evaluating the temporal variability of six benthic biological indicators;

and (ii) exploring the possible relationship between predictive abiotic and biotic responses variables in a temporal analysis. This work allows us to describe the behavior of abiotic and biotic variables during a long time period, in an area of remote coral reefs, where the effects of global changes should predominate over anthropogenic factors.

## MATERIALS AND METHODS

### Study area

The study was conducted at four coral reefs sites located in the south of the Guanahacabibes peninsula, Pinar del Rio Province, Cuba (Fig. 1). The sites, Cuevas de Pedro (CP) (21.8107 NL, 84.51169 WL); Yemayá (YE) (21.8345 NL, 84.4917 WL); Veral (VE) (21.9245 NL, 84.5419 WL); and Verraco (VR) (21.9166 NL, 84.6148 WL), were located on the reef upper edge next to the deep escarpment (fore reef), at depths between 13 and 15 m. The coral reefs of Guanahacabibes are close to the coast (less than 1 km), which is mostly karst with some areas with sandy beaches. The peninsula receives different degrees of chronic wind exposure, from the north, southeast and south direction (*Ballester, 1997*), and is also located in a zone frequently affected by hurricanes and tropical storms (*Perera-Valderrama et al., 2013*).

The coral reefs belong to the Guanahacabibes National Park, a marine protected area of 15,950 ha (*Márquez et al., 2013*). In the area, there is a small settlement (<80 inhabitants), and an international diving center that receives less than 6,000 divers per year (*Perera-Valderrama et al., 2017*). The area has no supplies of pollution (*e.g.*, cities, polluted rivers), and can therefore be considered as having a low level of human impact.

### Biological variables sampling methodology

The targeted coral reef sites were sampled, once a year during the summer season between 2008 and 2017 (except in 2009, when it was not possible to go out to sample), for a total of nine years, using biological variables from the Atlantic and Gulf Rapid Reef Assessment (AGRRA) protocol (*Kramer & Lang, 2003*). The sites were located in areas with continuous reefs with an extent of at least 200 m along corresponding isobaths. The distance between two adjacent sites was at least 5 km.

The coral sampling unit was a 10 m long linear transect (rope) and was used as a replicate within sites, with 15 transects sampled per site. We quantified living coral cover (%), which consists of the average percentage of live coral tissue intercepting the rope. We also quantified the number of colonies of scleractinian corals and hydrocoral species in each transect (below the transect line). A 25 cm square frame was used as a sampling unit to quantify fleshy algae cover (%), which included leafy, filamentous, globose and corticated macroalgae. Five frames, separated by 2 m, were placed along each transect. The average height (cm) of the fleshy algae per frame (measuring all algae) was also quantified.

### Acquisition of abiotic variables

The values of seven abiotic variables per site were obtained from remote sensing data sets (Table 1), considered possible drivers of structure and condition of coral reefs, as previously described in *Caballero-Aragón et al. (2022)*: sea wave height (WAVE), sea

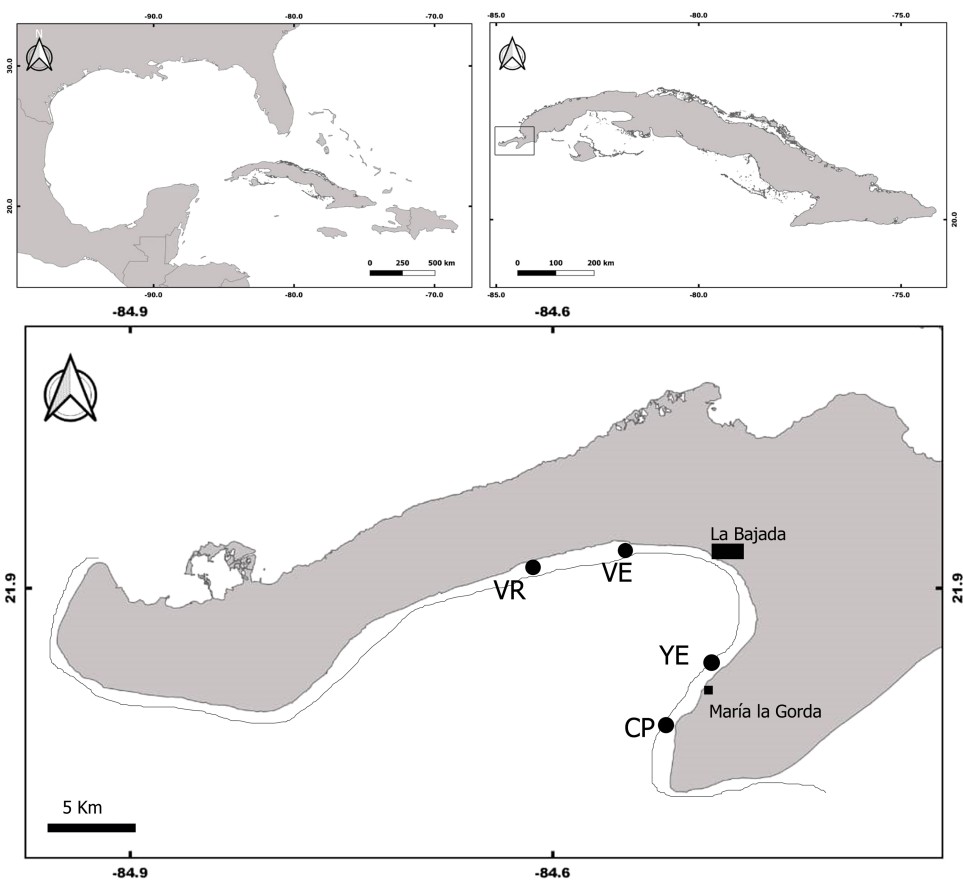

**Figure 1 Map of Guanahacabibes peninsula with the sampling sites.** VR, Verraco; VE, Veral; YE, Yemayá; CP, Cuevas de Pedro. Top panels refer to the Cuba position in the Great Caribbean region (left panel) and to the Guanacahabibes position inside the Cuba archipelago (right panel).

surface temperature (SST), degree heating week (DHW), chlorophyll-a concentration (CHL1), particulate organic carbon (POC), photosynthetically available radiation (PAR), and diffuse attenuation coefficient (KD490).

Sea wave height comes from GLOBAL_REANALYSIS_WAV_001_032 (https://resources.marine.copernicus.eu), and degree heating week from SIMAR (https://simar.conabio.gob.mx) (*Cerdeira-Estrada et al., 2019*). The other parameters belong to NASA Giovanni (https://giovanni.gsfc.nasa.gov) (*Acker & Leptoukh, 2007*). Each abiotic data presented different frequencies (daily, weekly or monthly) over a period of nine years between 2007 and 2016, excluding 2008 (Table 1).

## Data analysis

Living coral cover and abiotic variables, were represented using scatter plots graphs with the median (per year) as a measure of central tendency, and the mean of each site as a measure of variability (*Weissgerber et al., 2015*). The Fleshy Algae Index (FAI), coral species richness (S), and the Reef Functional Index (RFI), were presented using box plot graphs, with the mean as a measure of central tendency and confidential intervals as a measure of variability.
The FAI per site was calculated by averaging the results of multiplying the average of fleshy algae cover by its average height in each sampling unit. The coral species richness consisted of the number of coral species. To calculate the RFI, the criteria of *González-Barrios & Álvarez Filip (2018)* were followed, where they propose a functional index considering the morpho-functional attributes of each species or functional-coefficient (FC), according to the database of field studies in the Caribbean from the Healthy Reefs Initiative and other sources. The RFI per site was obtained from the sum of the product of the abundance (coverage per living coral species percentage) and the FC of each species (Table S1).

Following the same procedure as *Caballero-Aragón et al. (2022)*, for the row values of the abiotic variables described in Table 1, their mean and standard deviation were calculated, per year and for each site. Seven variables were obtained with the average condition (coded as AVE) and seven variables with the condition of variability (coded as SD), for a total of 14 abiotic variables to be related to the biological variables.

Coral species predominance and coral groups according to their morpho-functional traits, were visualized using a bar graph. A matrix of coral relative abundance (number of colonies per coral species divided by the total number of colonies) was constructed, including only those species that contributed to 95% of the total accumulated abundance, considering the remaining of species as rare. All coral species were classified into five morpho-functional groups according to the criteria of *González-Barrios & Álvarez Filip (2018)*: framework-building branching corals, massive reef framework-building corals, secondary massive reef framework-building corals, foliose-digitate species, and non-framework-building corals.

Differences in metrics were tested by a PERMANOVA analysis (*Anderson, Gorley & Clarke, 2008*), using a similarity matrix based on Euclidean distances for the univariate data and Bray Curtis for multivariate data, with 9,999 permutations and 0.05 significance. A two-way balanced design was applied, using "sites" and "years" as random factors. The magnitude of effects was assessed by the estimates of components of variation. Posterior pairwise test analysis was performed according to the presence or absence of the site-year interaction.

We used a distance-based linear model (DISTLM, *Anderson, Gorley & Clarke, 2008*) to explore the relationship between abiotic variables (combination of predictor variables) within the period 2007–2016 (excluding 2008), and biological variables (corals and algae) within the period 2008–2017, excluding 2008. Note that there is a lag of one year between the taking of abiotic data with respect to the taking of biological data. Our criterion was to relate the average of the abiotic variables of one year, with the values of the biological variables of the posterior year. We also included the FAI as a predictor variable for coral variables, because the coverage of the substrate by macroalgae can be considered as a driver for the coral community. Euclidean distance was used as the similarity index for univariate data, while the Bray-Curtis index was selected for multivariate data. DISTLM included the *forward* procedure method, using *the best-fitted model* by percentage of the explained variance (R2), performing 9,999 permutations.

Correlations between significant abiotic variables (obtained from DISTLM analysis) and living coral cover, FAI, S, and RFI, were visualized using scatter plots. The Distance-Based
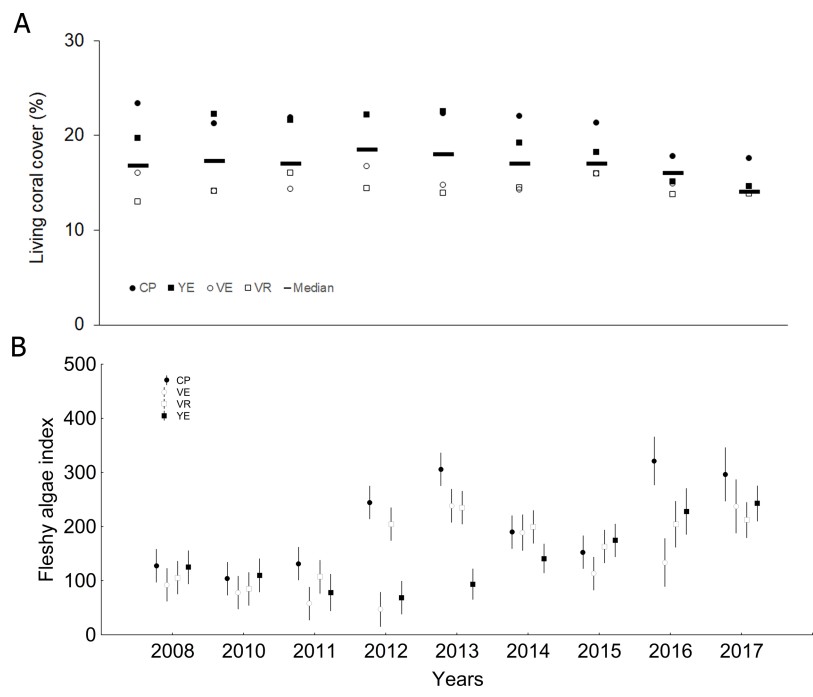

**Figure 2** Quantitative values per study year: (A) Living coral cover (bars = median; other symbols = mean per site), (B) Fleshy algae index (mean and confidential intervals per site). CP, Cuevas de Pedro; YE, Yemayá; VE, Veral, VR, Verraco.

Redundancy Analysis (dbRDA) was used to fit and visualize the results of DISTLM analysis for multivariate data (coral relative abundance, coral trait groups). Statistical procedures were performed using the PRIMER 6 + PERMANOVA program (*Clarke et al., 2014*).

## RESULTS

### Benthic community indicators

All coral community variables showed significant differences between sites, years, and the interaction sites x years, except for the living coral cover, which had no significant interaction (Table 2).

Living coral cover median (± SD) was 17.0 ± 6.9% (range: 13.8–23.3%), showing significantly lower values in the last two years compared to the previous ones (Fig. 2A, Tables S2 and S3). The fleshy algae index was 120.0 ± 123.2 (57.6–277.8), and great variability was found among years and sites (Fig. 2B, Tables S4 and S5). However, the general trend was towards an increase of mean values in 2016 and 2017.

Significant differences were observed among sites and years (Tables S6 and S7) with respect to coral relative abundance (species predominance), though four species conformed more than 65%: *Agaricia agaricites*, *Siderastrea siderea*, *Orbicella faveolata* and *Porites astreoides* (Fig. 3A). In the last two years, a certain general decrease in the predominance of *O. faveolata* and increase of *A. agaricites* were observed. The abundance of framework-building branching corals species was very low, and a tendency towards a decrease of

**Table 1  Abiotic variables taken from remote sensing, acronyms, description, source of data, and key references.**

| Abiotic variable/human impact proxy | Acronym | Product definition | Remote sensor and product calculation period | Key references |
|---|---|---|---|---|
| Sea wave height (m) | WAVE | Wave model calculating the frequency, direction and energy of waves. Spectrum that considers ocean currents and assimilates the height of significant waves based on historical altimetry records and satellite directional wave spectra. | MFWAM model, GLORYS12, Sentinel 1 SAR. Temporal resolution three hourly instantaneous at 20 km$^2$ from 2007 to 2016. | None[*] |
| Sea surface temperature (°C) | SST | Monthly nighttime temperatures of the sea surface, data derived from satellites infrared observations. | MODIS Global Mapped 11 μm Nighttime Sea Surface Temperature (NSST) from Terra and Aqua satellites, monthly at 4 km from 2007 to 2016. | *Kilpatrick et al. (2015)*[**] |
| Degree heating week (°C) | DHW | Sum of the weekly positive anomaly of the nocturnal sea surface temperature, compared to the monthly maximum of the adjusted climatology, over a 12 week period. | SATcoral-SIMAR product according to NOAA Coral Reef Watch methodologies using weekly data from OSTIA and GHRSST-MUR, at 1 km from 2007 to 2016. | *Cerdeira-Estrada et al. (2019)*[***] |
| Chlorophyll-a concentration (mg m$^{-3}$) | CHL1 | Chlorophyll-a concentration according to the GSM method, from the reflectance normalized to the original wavelengths of the sensor, without intercalibration. Variable widely used to estimate phytoplankton concentrations. | MODIS sensor on the AQUA satellite. Monthly data at 4 km from 2007 to 2016. | *O'Reilly et al (2000)*[**] |
| Particulate organic carbon (mol m$^{-3}$) | POC | Important component of the carbon cycle obtained from the original NASA algorithm (correlation of band proportions). It is considered a proxy for particulate organic matter. | MODIS sensor on the AQUA satellite. Monthly data at 4 km from 2007 to 2016. | *Stramski et al. (2008)*[**] |
| Photosynthetically available radiation (Einstein m$^{-2}$ day$^{-1}$) | PAR | Daily average of the photon flux density within the visible range (400–700 nm) of the light spectrum. Usable light in photosynthetic processes. | MODIS sensor on the AQUA satellite. Monthly data at 4 km from 2007 to 2016. | *Frouin, Franz & Werdell (2003)*[**] |
| Diffuse attenuation coefficient (m$^{-1}$) | KD490 | Based on the downwelling irradiance attenuation at 490 nm. Is obtained from the Morel algorithm. Considered a proxy of seawater turbidity. | MODIS sensor on the AQUA satellite. Monthly data at 4 km from 2007 to 2016. | *Morel et al. (2007)*[**] |

**Notes.**
[*] https://resources.marine.copernicus.eu.
[**] https://giovanni.gsfc.nasa.gov.
[***] https://simar.conabio.gob.mx.
[****] https://doi.org/10.1073/pnas.1708001115.

**Table 2 Results of the two-way PERMANOVA.** Analysis of the biological traits' variation considering sites and years. Significant p-values ($p < 0.05$) are given in bold. P (permutation $P$-value), df (the degrees of freedom), SS (sum of squares), CV (estimates of components of variation), perms (permutations).

| Variable | Source | df | SS | P(perm) | CV (%) | perms |
|---|---|---|---|---|---|---|
| Living coral cover | Site | 3 | 5171.7 | **0.0001** | 33 | 9953 |
| | Years | 9 | 849.9 | **0.0257** | 10 | 9940 |
| | Site × Years | 27 | 986.9 | 0.5090 | 3 | 9898 |
| | Residual | 566 | 21335 | | 57 | |
| Fleshy Algae Index | Site | 3 | 1030400 | **0.0001** | 13 | 9955 |
| | Years | 8 | 3595600 | **0.0001** | 21 | 9942 |
| | Site × Years | 24 | 2065700 | **0.0001** | 19 | 9913 |
| | Residual | 1420 | 15480000 | | 47 | |
| Coral richness | Site | 3 | 62.7 | **0.0069** | 14 | 9957 |
| | Years | 8 | 72.6 | **0.0063** | 11 | 9946 |
| | Site × Years | 24 | 99.2 | **0.0289** | 13 | 9896 |
| | Residual | 510 | 1285.5 | | 62 | |
| Coral relative abundance | Site | 3 | 24797 | **0.0005** | 10 | 9907 |
| | Years | 8 | 61474 | **0.0001** | 13 | 9883 |
| | Site × Years | 24 | 77539 | **0.0001** | 17 | 9738 |
| | Residual | 510 | 775300 | | 61 | |
| Relative abundance of coral traits groups | Site | 3 | 21006 | **0.0002** | 12 | 9922 |
| | Years | 8 | 41421 | **0.0012** | 14 | 9910 |
| | Site × Years | 24 | 51016 | **0.0001** | 19 | 9827 |
| | Residual | 512 | 395740 | | 55 | |
| Reef Funtional Index | Site | 3 | 0.13512 | **0.0003** | 20 | 9950 |
| | Years | 8 | 0.18678 | **0.0006** | 21 | 9933 |
| | Site × Years | 24 | 0.1012 | **0.0001** | 16 | 9895 |
| | Residual | 512 | 0.73815 | | 44 | |

framework-building massive coral species, and an increase of non-framework-building coral species and foliose-digitate coral species was observed (Fig. 3B, Tables S8 and S9). Coral richness was $17 \pm 3$ species (11–25 species), and a different pattern of variability among sites was observed. The largest number of species was found in 2017, mainly at the Cuevas de Pedro site (Fig. 4A, Tables S10 and S11). At the same time, RFI was $0.69 \pm 0.01$ (range: 0.67–0.73) with different pattern of variability among sites, but, observing a slight decrease of the index in the last two years (Fig. 4B, Tables S12 and S13).

## Abiotic variables

Spatial–temporal variability was observed in the abiotic variables (Table S12). Overall, the averages (AVE) of abiotic variables (except PAR) showed a higher median in the last year compared to the initial one. Regarding variability (SD), a similar trend was observed, although to a lesser degree (Fig. 5, Table S14).

## Relationship among variables/indicators

Significant results of DISTLM models are given in Table 3, and the scatterplots graphics are given in Fig. 6.

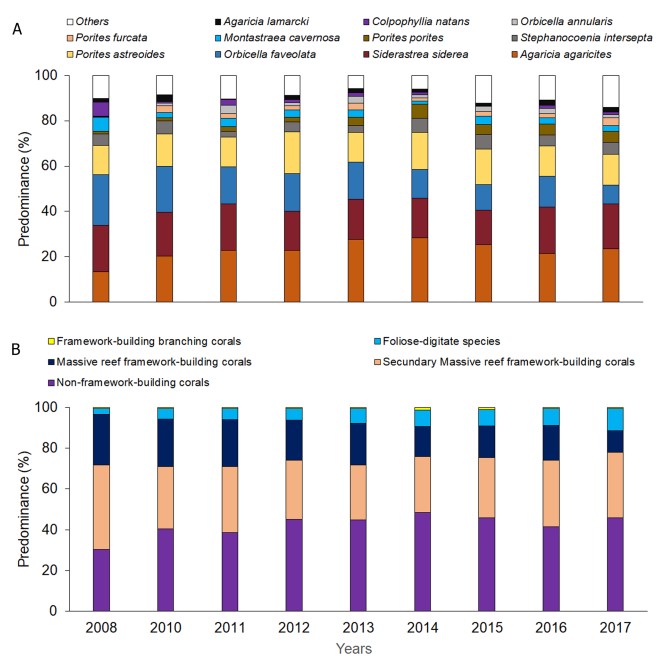

**Figure 3** Quantitative values per year: (A) coral relative abundance; (B) relative abundance of coral traits groups.

**Table 3   Results of the distance-based linear modeling (DISTLM), indicating the abiotic variables that best explain the similarity pattern of benthic community indicators on the basis of sequential tests (*p*-value < 0.05).** The full names of abiotic variables are given in Table 1. AVE: average; SD: standard deviation; R2: coefficient of determination; P: permutation *P*-value.

| Coral community variable (response) | Abiotic variable (predictor) | R² individual | Cumulative explained variance (%) | *p*-value |
|---|---|---|---|---|
| Living coral cover | AVE PAR | 0.19 | 19 | **0.008** |
| | AVE DHW | 0.16 | 35 | **0.008** |
| | SD CHL1 | 0.08 | 42 | **0.048** |
| Coral richness | AVE DHW | 0.11 | 11 | **0.042** |
| | AVE WAVE | 0.11 | 21 | **0.031** |
| | AVE KD490 | 0.10 | 33 | **0.027** |
| Reef funtional index | AVE NSST | 0.26 | 23 | **0.003** |
| | SD KD490 | 0.23 | 49 | **0.001** |
| Fleshy algae index | AVE NSST | 0.17 | 17 | **0.012** |
| | SD NSST | 0.17 | 35 | **0.006** |
| Coral relative abundance | SD KD490 | 0.08 | 8 | **0.008** |
| | AVE NSST | 0.07 | 15 | **0.011** |
| Coral trait groups relative abundance | Fleshy algae index | 0.17 | 17 | **0.001** |
| | SD KD490 | 0.10 | 27 | **0.008** |

Three variables significantly explained the 42% of the variability of living coral cover (Table 3): AVE-PAR, AVE-DHW and SD-CHL1; AVE-PAR positively influenced the

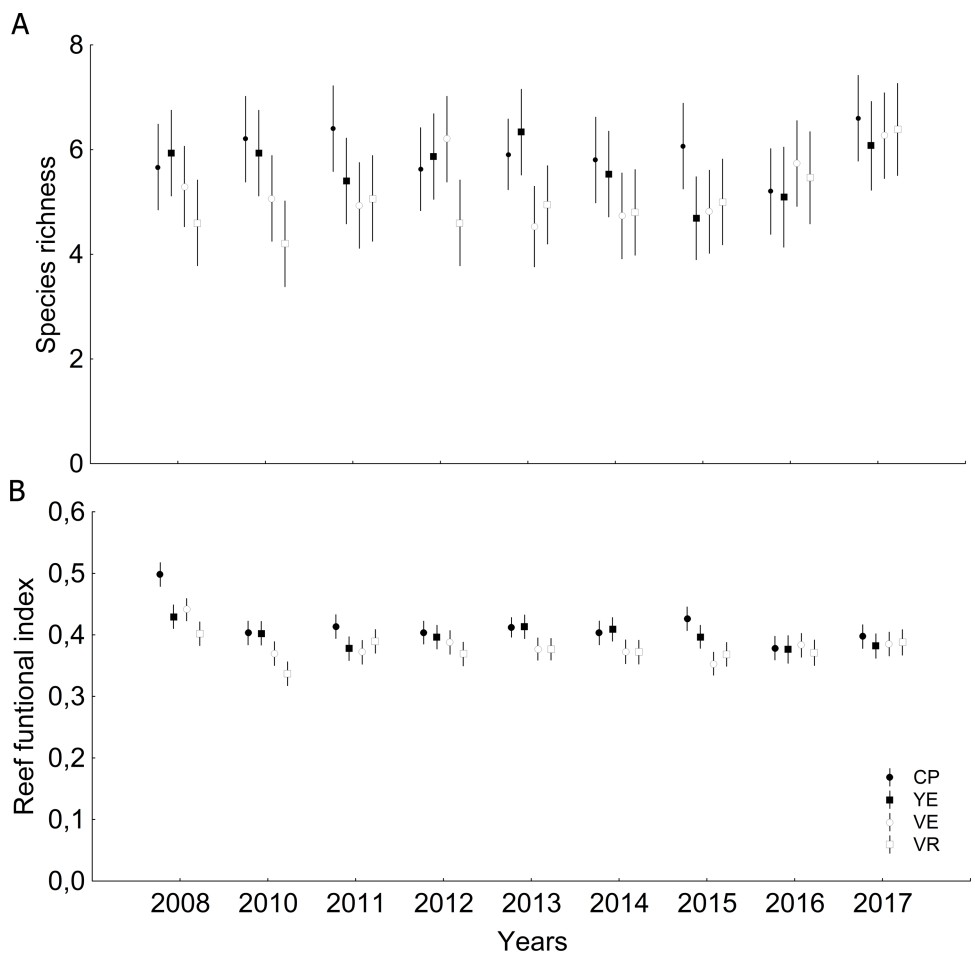

**Figure 4** **Quantitative values (mean and confidential intervals per site): (A) Species richness; (B) Reef functional index.** CP, Cuevas de Pedro; YE, Yemayá; VE, Veral; VR, Verraco.

coral cover while the other two negatively (Figs. 6A–6C). AVE-DHW, AVE-WAVE and AVE-KD490 explained the 33% of the variability of coral richness; the three variables positively influenced the biological variable (Fig. 6D–6F). AVE-NSST and SD-KD490 explained the 49% of the variability of RFI, negatively influencing it (Figs. 6G and 6H). Two variables significantly explained the 35% of the variability of FAI: AVE-NSST and SD-NSST; the first influenced positively while the second negatively (Figs. 6I and 6J).

SD-KD490 and AVE-NSST explained the 15% of the variability of coral relative abundance, while, FAI and SD-KD490, explained the 27% of the variability of the relative abundance of coral trait groups. The sites had great dispersion, and the abiotic and biological variables vectors did not show a clear interpretation in relation to the dispersion of the sites, according to dbRDA model. There is a certain trend in the vectorial direction of the variables AVE-NSST and SD KD490 towards the last years of sampling, coinciding with the inverse direction of the relative abundance vector of *O. faveolata* (Figs. 7A and 7B). Similarly, towards the last year of the study, a certain coincidence is observed in the

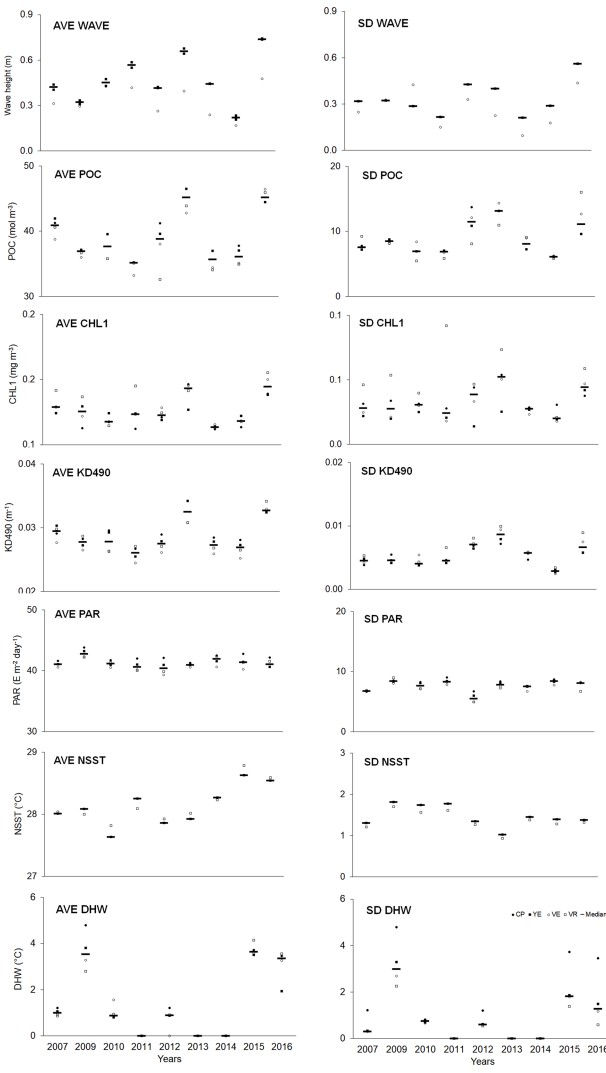

**Figure 5** **Quantitative values (bars = median; other symbols = mean per site) of abiotic variables.** The full names of the abiotic variables are given in Table 1. CP, Cuevas de Pedro; YE, Yemayá; VE, Veral; VR, Verraco.

direction of the vectors FAI and non-framework-building corals species (Figs. 7C and 7D).

## DISCUSSION

Previous studies in the coral reefs of Guanahacabibes have recorded recent mortality in massive reef-building corals, due to disease outbreaks (white plague) and bleaching events in the summer months (*Alcolado et al., 2003*; *Guardia, Valdivia & González-Díaz, 2004*; *Caballero-Aragón et al., 2007*). Similarly, the area was affected by Hurricane Ivan in 2004, which was a category five on the Saffir-Simpson scale (https://www.nhc.noaa.gov/). Hurricane impacts at individual sites can reduce coral cover, change species composition, and affect the functioning of coral communities (*Hoegh-Guldberg et al., 2007*; *Kennedy*

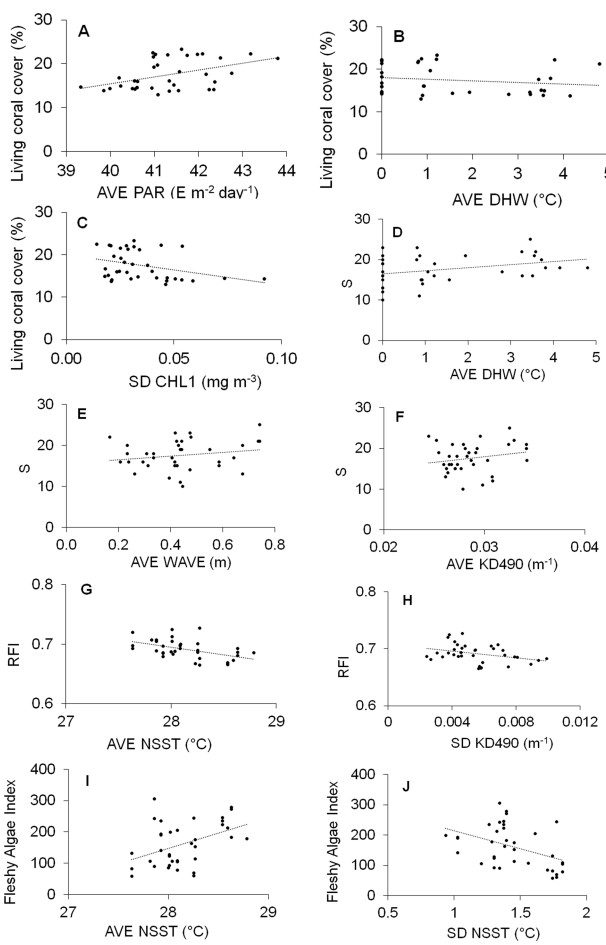

**Figure 6** Scatter plot graphs between abiotic predictor variables indicated as significant by the DIS-TLM routine and response indicators. Abiotic full name in Table 1. S, Richness; RFI, Reef functional index; FAI, Fleshy algae index; AVE, average; SD, standard deviation.

*et al., 2013*). However, the causes of the decline of living coral cover observed after 2015 are instead not clear. No hurricanes occurred during this period, and no increase in anthropogenic activities (tourism, agriculture or coastal development) was reported on the peninsula. On the other hand, diseases have been observed in isolation and bleaching events occur every summer with different intensities (paper in preparation).

In contrast to Guanahacabibes, clear indicators of anthropogenic impacts have also been observed in Caribbean and Florida coral reefs, including slow coral growth rate, coral mortality due to coral bleaching and virulent diseases, decreased complexity of coral reefs, low diversity of coral species and low population density, high density of filamentous and fleshy algae, high density of sponges, and coral colonies covered by fine sediments (*Weil, 2004*; *Jackson et al., 2014*; *Lapointe et al., 2019*; *Sánchez et al., 2019*). In turn, they contrast with recent observations in the Mesoamerican Reef (MAR) region, where the coral cover has remained relatively stable (*Suchley, McField & Alvarez-Filip, 2016*; *Suchley & Álvarez Filip, 2018*; *McField et al., 2018*).

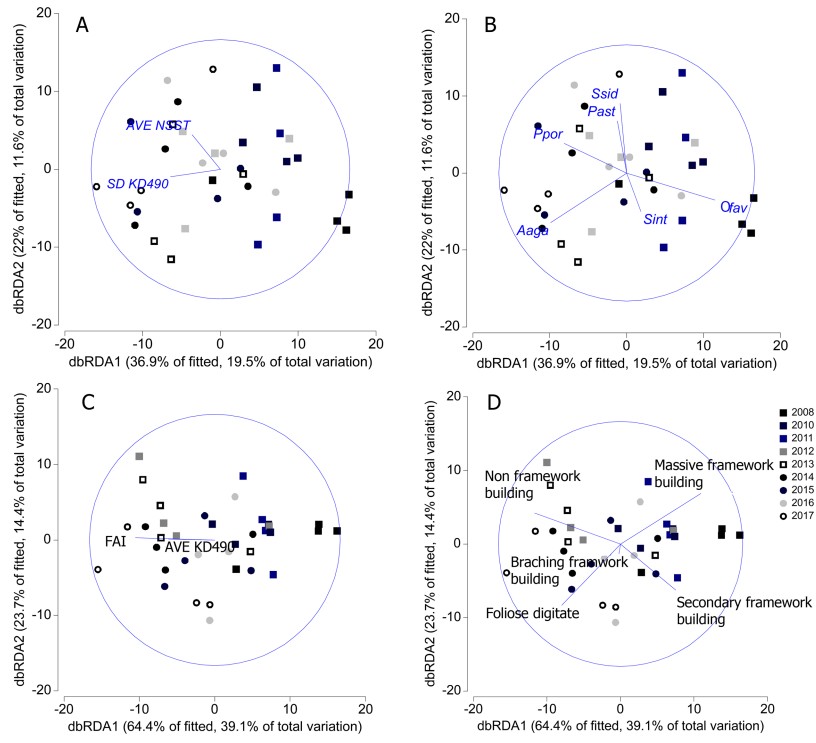

**Figure 7 Distance-based redundancy analysis (dbRDA) relating significant abiotic variables to the similarity patterns of benthic reef communities.** In A and C, vectors indicate the abiotic variables (full names in Table 1). In B and D, vectors indicate the most important coral species and coral trait groups. Aaga, *Agaricia agaricites*; Ofav, *Orbicella faveolata*; Past, *Porites astreoides*; Ppor, *P. porites*; Sint, *Stephanocoenia intercepta*; Ssid, *Siderastrea siderea*. FAI, Fleshy algae index; AVE, average; SD, standard deviation.

Coral decline in the Caribbean has been accompanied by a rapid increase in macroalgae (*Hughes et al., 2007*; *Paddack, Reynolds & Côté, 2009*). *Suchley, McField & Alvarez-Filip (2016)* reported a rapid increase in fleshy cover in the Mexican Caribbean coral reefs since 2005. This is largely attributed to coastal landscape transformation, induced by the development of tourism infrastructure (hotels and restaurants) and cruise ship ports, associated with significant eutrophication and water pollution resulting from inadequate wastewater treatment (*Hernández-Terrones et al., 2015*; *Martínez-Rendis et al., 2015*; *Arias-González et al., 2017*; *Rioja-Nieto & Álvarez Filip, 2018*). Biomass loss of herbivorous fishes, is also strongly associated with habitat degradation and human activities such as coastal development and fishing intensity (*Alvarez-Filip et al., 2015*; *Valdivia, Cox & Bruno, 2017*).

In Guanahacabibes, anthropogenic disturbances do not seem to be the main cause of the tendency to increase fleshy algae. However, a possible source of pollution, somewhat distant (more than 25 km), could be the port of La Coloma (*Caballero & Perera, 2014*). On some occasions, we observed greenish water masses (apparently rich in nutrients), coming from the west of the Gulf of Batabanó; the source of the organic matter seemed to be represented by La Coloma River, where the town of La Coloma is located. However,

we have no quantifiable evidence, of the effect or increase of these events, and we have no evidence of changes in the current dynamic, which could accentuate their effect.

Regarding herbivorous fish populations, we do not have a complete quantitative analysis for this study (paper in preparation). The most recent report on herbivorous fish populations (*Cobián-Rojas et al., 2011*) indicated an average density and biomass of 0.8 individuals/m$^2$ and 23 g/m$^2$, respectively, with a predominance of medium-sized specimens of the Scaridae family and conglomerates of the Acanthuridae family. Although we did not perform a quantitative data analysis, based on these data, we can infer that the situation of herbivorous fishes is not critical for the reef, but we cannot define whether this influences the increasing trend of fleshy algae. The opposite occurs with the populations of black sea urchins *Diadema antillarum*, which practically were not observed during the day in the fore reef, being very abundant in the shallow coastal zone (1 m depth) of the entire southern part of the peninsula (*Caballero & Perera, 2014*). The lack of black urchin observations in the Guanahacabibes fore reefs could have implications for algal control.

The four dominant coral species of Guanahacabibes reefs are the same ones that dominate most of the fore reefs of Cuba (*Caballero-Aragón et al., 2019*). The decrease of *O. faveolata* abundance, and increase of *A. agaricites* may be closely related to the declining trend of massive reef-building species and RFI, and it is interesting to observe, an increase in species richness but a decrease in functional diversity. Our hypothesis is that the decrease in the predominance of *O. faveolata* has favored the increase in the number of opportunistic species that are more resistant to environmental disturbances.

In Caribbean reefs, it has been shown that a large proportion of sites are dominated by secondary coral species (*e.g.*, weedy corals), decreasing their resilience, and favoring the increase of algal cover and other sessile organisms (*Bruno, Côté & Toth, 2019*; *González-Barrios, Cabral Tena & Álvarez Filip, 2018*). The morpho-functional traits of species provide general and predictable rules for understanding the dynamics of ecological communities (*González-Barrios & Álvarez Filip, 2018*). These authors found for the MAR, that most species contribute little to the site-level reef-building potential, such as the widely spread *Agaricia* spp. and *Porites* spp. Conversely, species with high-functional potential such as *Orbicella* spp and *Acropora* spp have limited relative abundance and distribution. Several studies also report that non-framework species are rapidly increasing their relative abundance in the Caribbean (*Green, Edmunds & Carpenter, 2008*; *Jackson et al., 2014*; *Perry et al., 2015*).

The decreasing dominance of branching-massive framework-building coral species has serious consequences for reef functioning, being compromised in the future, with negative calcium carbonate budgets and low structural complexity (*Kennedy et al., 2013*; *Perry et al., 2015*; *González-Barrios & Álvarez Filip, 2018*). Reef functioning increases with coral cover, but the magnitude of functional increase depends on the composition and dominance patterns of key groups on the reef; a strong correlation between RFI and the living coral cover was found for 170 sites along the MAR (*González-Barrios & Álvarez Filip, 2018*). In our temporal analysis, we found a positive correlation between branching-massive framework-building coral species and living coral cover; as well as found a positive correlation between RFI and living coral cover (Fig. S1). Therefore, we could infer that the

decrease in living coral cover values responds to the decline in the dominance of *Orbicella* species.

The results of the DISTLM analysis found significance the relationship between some abiotic and biological variables, however, in none of the cases, the coefficients of determination were high, which is interpreted as a weak association between the variables. Scatterplot graphs also did not show a clear association between the variables, observing a large dispersion. We consider this objective would be only "exploratory", and the results should be taken with certain measures.

An increase in thermal anomalies could increase the deterioration in condition and abundance of reef-building corals such as *Acropora* or *Orbicella* species. Heat stress may cause coral bleaching, disease outbreaks, and subsequent coral mortality, also increasing corals vulnerability to other natural or anthropogenic stressors (*Hughes et al., 2017*; *Hughes et al., 2018a*; *Hughes et al., 2018b*; *Muñiz Castillo et al., 2019*). Heat stress has been considered a driver with a negative effect on the living coral cover of Australian reefs, with $R^2$ values of 0.39 (*Ceccarelli et al., 2019*). Nonetheless, AVE DHW seems to favor the temporal increase of species richness, and this also seems reasonable. We found an inverse relationship between the *Orbicella* species abundance and the general species richness in the reef. We found the tendency to increase of "weedy species" more tolerant to disturbances, such as thermal anomalies (*Barranco et al., 2016*; *Cramer et al., 2021*).

The increase in sea temperature may favor the growth of some fleshy algae species (*McClanahan, Muthiga & Mangi, 2001*; *Ateweberhan, Bruggemann & Breeman, 2006*; *Rashedy et al., 2022*); however, the increase of fleshy algae on the reef is mainly associated with the scarcity of herbivores or with the increase of nutrients (*Hernández-Terrones et al., 2015*; *Martínez-Rendis et al., 2015*; *Suchley, McField & Alvarez-Filip, 2016*). In our study, the temporal variability of the abiotic variables (proxies) associated with an increase in nutrients (POC, CHL1 or KD490), did not show a temporal relationship with FAI. AVE NSST and SD KD490 showed a negative relationship with RFI and had the highest regression coefficients (0.26 and 0.23); however, we did not find a clear explanation for this temporary behavior.

*Caballero-Aragón et al. (2022)*, studied the relationship between abiotic and biotic variables in 73 Cuban coral reefs, where WAVE and NSST were identified as drivers. Conversely, in our temporal analysis, no significant relationship was found between the WAVE and the biological variables, although the former tended to increase over time. Apparently, since no severe meteorological events (high-intensity hurricanes) occurred during the study period, the slight increase in WAVE during the period was not enough to cause significant effects on biological variables.

The weak association between abiotic variables (proxies of anthropic disturbances) and biological variables, is probably associated with the relative homogeneity and low values observed. By "low values", we refer, for example, to the average of CHL1 of 0.12 mg m-3, with maxima of 0.16 mg m that we found in Guanahacabibes. This same variable was reported in the Florida Keys with averages of 0.25 mg m$^{-3}$ and maxima of 1.25 mg m$^{-3}$ (*Lapointe et al., 2019*). Another example of "low values" is KD490, a variable that provides an estimate of turbidity (*Ceccarelli et al., 2019*). Our KD490 values did not reach 0.04, and

in some coral reefs in Puerto Rico values of up to 0.51 were reported (*Hernández-Delgado & Ortiz-Flores, 2022*). In the Abrolhos complex, a group of reefs closer to the coast, a Kd490 average of 0.11 was detected; in the same area, but in another reef, about 60 km far from the coast, the authors registered a Kd490 of 0.08 (*Freitas et al., 2019*). *Hernández-Delgado & Ortiz-Flores (2022)* also reported CHL1 values from up to 13.3 mg m$^{-3}$ and POC values of 2481 mol m$^{-3}$. POC is another variable proxy for seawater nutrient concentrations (*Stramski et al., 2008*), and in Guanahacabibes, our values did not reach 50 mol m$^{-3}$.

## CONCLUSIONS

The results may corroborate our hypothesis that Guanahacabibes is a "remote reef with low human impact", where acute anthropogenic effects are not clearly observed. Two different trends were observed: a slight decrease in living coral cover in the last years, and an increase in the fleshy algae index. Likewise, a decrease in the functionality of the reef is appreciated, taking into account the decrease in the predominance of branching-massive framework reef-building species, with a significant decline in the predominance of *Orbicella* species. Nevertheless, a weak association between abiotic and biological variables was observed in the temporal analysis. The current scenario of the condition of the coral reef seems to be regulated by the global effects of climate change, with less acute and longer-term effects.

## ACKNOWLEDGEMENTS

We thank to the Director of the Guanahacabibes National Park, the National Aquarium of Cuba, Maria la Gorda International Diving Center and the National Center of Protected Areas.

### Funding

This work was supported by UNDP-GEF projects "Enhancing the Prevention, Control and Management of Invasive Alien Species in Vulnerable Ecosystems" and "Application of a Regional Approach to the Management of Marine and Coastal Protected Areas in Cuba's Southern Archipelagos Region". The funders had no role in study design, data collection and analysis, decision to publish, or preparation of the manuscript.

### Grant Disclosures

The following grant information was disclosed by the authors:
Enhancing the Prevention, Control and Management of Invasive Alien Species in Vulnerable Ecosystems.
Application of a Regional Approach to the Management of Marine and Coastal Protected Areas in Cuba's Southern Archipelagos Region.

### Competing Interests

The authors declare there are no competing interests.

## Author Contributions

- Hansel Caballero-Aragón conceived and designed the experiments, performed the experiments, analyzed the data, prepared figures and/or tables, authored or reviewed drafts of the article, and approved the final draft.
- Susana Perera-Valderrama conceived and designed the experiments, performed the experiments, analyzed the data, prepared figures and/or tables, authored or reviewed drafts of the article, and approved the final draft.
- Dorka Cobián-Rojas performed the experiments, authored or reviewed drafts of the article, and approved the final draft.
- Zaimiuri Hernández Gonzalez performed the experiments, authored or reviewed drafts of the article, and approved the final draft.
- Juliett González Méndez performed the experiments, authored or reviewed drafts of the article, and approved the final draft.
- Elena De la Guardia conceived and designed the experiments, performed the experiments, analyzed the data, prepared figures and/or tables, authored or reviewed drafts of the article, and approved the final draft.

## Data Availability

The raw data are available in the Supplementary File.

## Supplemental Information

Supplemental information for this article can be found online at http://dx.doi.org/10.7717/peerj.15953#supplemental-information.

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
