# Peer review of "A decade of study on the condition of western Cuban coral reefs, with low human impact"

_PeerJ, doi:10.7717/peerj.15953_

## Round 0.1 · original submission · Major Revisions

All the referees agreed that your study is of crucial importance to document the long-term effects of climate change, but they also gave suggestions to improve the quality of the manuscript. Therefore, I suggest major revisions.

Reviewer 1 ·

Basic reporting

Overall, the manuscript in clear and easily understandable. Nonetheless, an English revision of the manuscript from a professional or a fluent English speaker is suggested.
The cited literature is adequate to the topic.

Experimental design

The study is well designed and fit the aims of the journal.
Materials and methods are clearly presented and explained.

Validity of the findings

The results of this long-term study are of crucial importance in the context of the current climate crisis. Raw data and supplementary files are also shared helping in the completeness and understanding of the manuscript.

Annotated reviews are not available for download in order to protect the identity of reviewers who chose to remain anonymous.

Reviewer 2 ·

Basic reporting

Generally, the article is well written, with professional English language; nevertheless some sentences should be rewritten, resulting ambiguous, not clear or not correct (e.g. lines 19, 78, 142, 211). Please, also check the tables and figures numeration.

Experimental design

The article is original enough, although it would be appropriate to highlight which are the main innovations, with respect to not sufficiently mentioned literature on biological indicators and indices.
Methods are well described, except the biological variables that result too briefs; indeed, they refer to the AGRRA protocol, but a more exhaustive description in the text is needed. Moreover, adding some example species, later mentioned in the discussion, may result appropriate.
I just not well understand the choice to relate abiotic and biotic variables with a shift of one year (line 178). Some biological responses to stress have relatively short times; maybe, could be useful have more relation times, for example time 0, after 6 months and 1 years.

Validity of the findings

Discussion and conclusions are relevant and well linked to the findings.
Please, pay attention to the interpretation of Distlm results and the assertion "Some abiotic variables significantly explained the variability of biological indicators, however, determination coefficients were low", where the R2 low values do not explain a significant relationship among biotic and abiotic variables.

Reviewer 3 ·

Basic reporting

The article is written in relatively fluent English, although there are several instances where there is some ambiguity and minor grammatical errors (see marked up pdf). The literature references are appropriate, and provide significant background. However, there are some missing references, links that do not link back to the references, and some areas where the citation information is not correct (e.g., linking to the product ID, rather than the product name). The use of acronyms makes some parts of the manuscript difficult to follow, but these could be solved through more judicious use of tables and acronyms.
The literature references are appropriate, though there is at least one that is either missing or mislabeled, and the authors need to check the links they supply - one of the links to a manual actually leads to an adult website.
The figures are quite clear, but have some shortcomings (see marked up pdf). Amongst these, is the use of averages for sites, when I suspect the authors mean 'mean' and the lack of indication of variability within sites. Confidence intervals (95%CI, SE or SD) should be given around each mean. Further, the mean is also a measure of central tendency, and not a measure of variability as the authors refer to it. I believe that some of the figures, particularly the non-significant abiotic variables, could be places in SI, and that some more care is needed in producing some of the figures (e.g., the map of the sites). See marked up pdf.
The introduction is short, and is only a very brief description of the background. It would be good to see here some discussion of the abiotic variables that will be investigated, the rationale in choosing them, and the expected relationships between these and the different benthic categories collected. The results are relevant, but they appear to have been over-interpreted (see below).
The discussion, as it stands is over long and I believe most of this is the authors trying to fit expected results to non-significant patterns in the data. By moving some rationale and expected outcomes to the introduction (eg lines 303-323), and commenting on the results, as opposed to non-existent trends or potential variables not investigated (such as herbivorous fish), this would help tighten and shorten the discussion considerably.

Experimental design

Research question: the rationale behind why certain individual abiotic variables and the benthic variables were chosen is not well developed. While looking at global influences, it does make obvious sense to include degree heating weeks and SST, the rationale behind waves, turbidity and PAR (which are likely influenced by local conditions) is less clear. I think this is to control somewhat for local conditions, but this is not explicitly stated, and the lack of a control site in a more impacted area limits the conclusions that can be drawn from this experimental design. Overall, if confined to discussing the patterns seen o

Methods: there are some areas of methods which need expanded upon, and the data analysis is described in great (almost too great) detail, but still misses some important information, particularly about post-hoc tests.

Validity of the findings

All data have been provided. However, I am not an expert in PERMANOVA or the use of PRIMER, and I haven't been able to assess that the statistical tests done on these are sound (though in my limited knowledge, I cannot find an obvious fault). However, the post-hoc pairwise tests need to be described more clearly in the methods, and some account needs to be taken for the repeated tests and the likelihood of type 1 errors (e.g. a Bonferroni correction). I also believe that more work should be done by the authors on potential differences between sites. For example, the authors in the discussion state that across Cuba, there is an effect of Wave exposure on live coral cover, but despite increasing wave energy this was not observed here. That may be the case overall, but at least two of the sites appear more exposed, and experience greater wave energy while showing larger declines in coral cover (Cuevas de Pedro and Yemay) than the other two.

The conclusions are not clearly stated or linked to the original research question (because this question, and the hypotheses were not made clearly in the introduction). Further, the authors bring in data that is yet to be published, make quite speculative and broad comments about tends at both their sites and in the wider Caribbean. For example, while there is no significant statistical result for differences between years in live coral cover from 2008 - 2015, the authors state that there is an increasing and a decreasing trend within these years. That conclusion cannot be drawn from the results. There are several points in the discussion where results and potential patterns are discussed without clear connections to the results (such as when discussing herbivore fish biomass). On that last point, considering that herbivore activity is important on Caribbean reefs, and the authors state that there is a study in preparation on that topic, I would suggest including such biotic variables into this analysis.

Additional comments

The manuscript, although promising requires considerable revision. This should include more strongly placing the study in context, giving rationale behind the choice of abiotic variables chosen to test, the inclusion of an important biotic variable as a potential driver (herbivorous fish biomass), and greater clarity in how post-hoc tests were conducted and the variation in the data. The discussion currently is overlong, and somewhat confused that too often describes non-significant trends, or diverts into a discussion about wider Caribbean trends without refocusing back to the study area.

Annotated reviews are not available for download in order to protect the identity of reviewers who chose to remain anonymous.

---

## Round 0.2 · Minor Revisions

The authors met all the issues raised by referee who appreciated the improved versione of the paper. There are a few and customary changes that could improve the manuscript, therefore, I suggest to considerate the new comments by referee 1.

Reviewer 1 ·

Basic reporting

Authors have addressed the suggested comments of both reviewers.
The readiness of the manuscript increased significantly compared to the first version.

Experimental design

The experimental design was fine already in the first version.

Validity of the findings

This long-term study is of crucial importance for a better understanding of the processing occurring in "isolated" reefs

Annotated reviews are not available for download in order to protect the identity of reviewers who chose to remain anonymous.

---

## Round 0.3 · Minor Revisions

The authors aimed to describe the changes of abiotic and biotic variables during a long time period, in an area of remote coral reefs, where the effects of global changes should predominate over anthropogenic factors; as such a fixed statistical design could be acceptable.

However, referee 1 suggested a revision in the statistical design and considering "sites" and "years" as random factors in order to give a wider/global perspective to this paper. “

If you don’t agree with this change please give a clear justification to the reviewer.

Reviewer 1 ·

Basic reporting

The overall study is clear and well structured, and the authors really improved the written English, making the manuscript fluent and easily readable.

Experimental design

The experimental design is correct, just the two considered factors "site" and "year" should be random not fixed. This must be corrected by the authors.
Please see the attachment file.

Validity of the findings

In the context of the current climate crisis, long-term studies as the one presented here are of fundamental importance to understand and diagnose changes in the coral reef status.

Annotated reviews are not available for download in order to protect the identity of reviewers who chose to remain anonymous.

---

## Round 0.4 · accepted · Accept

The authors met the suggestions made by referee 1 and changed the statistical design as suggested, so the paper is now acceptable for publication.